# Challenges and Opportunities for High-Power and High-Frequency AlGaN/GaN High-Electron-Mobility Transistor (HEMT) Applications: A Review

**DOI:** 10.3390/mi13122133

**Published:** 2022-12-01

**Authors:** Muhaimin Haziq, Shaili Falina, Asrulnizam Abd Manaf, Hiroshi Kawarada, Mohd Syamsul

**Affiliations:** 1Institute of Nano Optoelectronics Research and Technology (INOR), Universiti Sains Malaysia, Sains@USM, Bayan Lepas 11900, Pulau Pinang, Malaysia; 2Collaborative Microelectronic Design Excellence Center (CEDEC), Universiti Sains Malaysia, Sains@USM, Bayan Lepas 11900, Pulau Pinang, Malaysia; 3Faculty of Science and Engineering, Waseda University, Tokyo 169-8555, Japan; 4The Kagami Memorial Laboratory for Materials Science and Technology, Waseda University, 2-8-26 Nishiwaseda, Shinjuku, Tokyo 169-0051, Japan

**Keywords:** HEMTs, 2DEG, GaN, challenges, opportunities

## Abstract

The emergence of gallium nitride high-electron-mobility transistor (GaN HEMT) devices has the potential to deliver high power and high frequency with performances surpassing mainstream silicon and other advanced semiconductor field-effect transistor (FET) technologies. Nevertheless, HEMT devices suffer from certain parasitic and reliability concerns that limit their performance. This paper aims to review the latest experimental evidence regarding HEMT technologies on the parasitic issues that affect aluminum gallium nitride (AlGaN)/GaN HEMTs. The first part of this review provides a brief introduction to AlGaN/GaN HEMT technologies, and the second part outlines the challenges often faced during HEMT fabrication, such as normally-on operation, self-heating effects, current collapse, peak electric field distribution, gate leakages, and high ohmic contact resistance. Finally, a number of effective approaches to enhancing the device’s performance are addressed.

## 1. Introduction

The gallium nitride high-electron-mobility transistor (GaN HEMT) has attracted the interest of many researchers as a power device platform due to its high operating frequency, high breakdown voltage, high-temperature capability, reduced on-state resistance, and high electron saturation velocity [1]. GaN-based HEMTs can produce high current densities and low channel resistances due to their high electron mobility and high carrier concentration [2]. These merits are vital for a wide range of high-power and high-frequency applications [3], especially in the areas of communications, radar, and space [4,5]. However, various limitations hinder GaN HEMTs’ ability to be fully commercially exploited [6], including reliability and short-channel effect concerns. Moreover, their dynamic on-state resistance (R_DS,ON_) values worsen during high-voltage switching, which wastes energy and compromises the system’s reliability. This deterioration is often induced by a phenomenon known as current collapse, which is triggered by charge trapping by surface states in the drift region and bulk traps in the buffer layers [7]. Furthermore, as the size of transistors has decreased over time in efforts to boost their performance speed, scaling has become more difficult due to self-heating issues.

Much research has been conducted over the years to highlight the influence of a component’s structural behavior on its electrical properties to assess its reliability [8]. However, based on past achievements in improving aluminum gallium nitride (AlGaN)/GaN HEMT performance, only a few address the multiple challenges associated with HEMT devices. Most studies focus on one specific issue, such as self-heating, with limited exposure to some of the recent structural improvements. To address this gap, our study summarizes practical and updated approaches over the past six years (from 2017 to the present) to counter the various HEMT challenges at the device level, such as normally-off operation, self-heating, current collapse, peak electric field distribution, gate leakage, and high ohmic contact resistance. Aside from that, this review offers immense opportunities for optimum HEMT device performance. Section 2 provides a brief introduction to the conventional AlGaN/GaN HEMT technology, mainly focusing on its design and operating principles. Section 3.1 discusses the normally-on mode of traditional HEMT devices and several practical approaches to achieving normally-off operation. Further, we present different HEMT substrates to improve the device’s thermal management (Section 3.2). In Section 3.3, field-plate (FP) implementation, surface passivation, and different gate structures are introduced to overcome current collapse, peak electric field distribution, and gate leakage issues. In Section 3.4, the impact of varying metal contact combinations to surpass the high ohmic contact resistance is explained. Finally, Section 4 provides an overall summary of the review.

## 2. AlGaN/GaN HEMT Technology

AlGaN/GaN HEMT technology offers exceptional current density and output power devices and is poised to become the dominant technology for various applications [9]. Due to their broad energy band gap, high critical electric field, and solid thermal dissipation capabilities, substantial research has been aimed at developing III-nitride compound semiconductors for optoelectronic and electronic devices [10,11]. The inherent physical features of nitride-based semiconductors allow for high off-state voltage, low on-state resistance, and high-power density [12,13]. Recent studies have shown that, within power device applications, GaN-based devices are superior to gallium arsenide (GaAs)-based devices [14], with the former promising greater input power robustness [15,16]. Moreover, GaN has a saturation electron velocity (v_sat_) two or more times faster than silicon (Si) and GaAs, with a dielectric field strength (E_c_) ten times greater than Si and 7.5 times greater than GaAs [17]. Consequently, GaN-based HEMTs are functionally superior to Si-based HEMTs, as they provide a higher operating frequency, output power, and operating temperatures [18]. Table 1 lists the material properties of AlGaN and GaN recently implemented in HEMT design [19,20,21,22,23,24,25,26,27,28,29,30].

As observed in Table 1, many reports of high Al-composition AlGaN channel devices have surfaced in recent years due to better critical breakdown and electric field distribution [31,32,33], allowing for exceptionally high voltage transistor operations. In addition, the particular contact resistivity degrades as the Al concentration increases [34], which is very appealing for the development of ultra-wide bandgap (UWBG) semiconductor devices because of the inherent polar nature of III-nitride materials and the availability of high thermal conductivity substrates [35]. In terms of radio frequency (RF) performance, an AlGaN-channel HEMT has exhibited the highest documented current gain cut-off frequency (f_T_) at 40 GHz [36].

A high degree of reliability is necessary for GaN power devices to be employed in critical electronic systems such as HEMTs, considering the failure mechanisms associated with high-voltage and high-temperature applications [37]. The typical configuration of AlGaN/GaN HEMT consists of a heterojunction formation following the alignment of adjacent wide and narrow band gap semiconductors, as shown in Figure 1. In the AlGaN barrier layer, the gate metal produces a Schottky contact, which controls the polarization charge density at the AlGaN/GaN interface. All the electrons then temporarily cluster near AlGaN due to the material’s higher band gap, which acts as a barrier. As a result, an impenetrable two-dimensional electron gas (2DEG) forms directly in GaN, near the AlGaN border. Many factors influence the quality of the 2DEG, including substrate materials, growing method, and the doping level of the carrier supply layer. This 2DEG formation [38,39,40] allows power electronics engineers to construct devices with increased efficiency and power density while lowering costs [41].

Unlike GaAs-based devices, GaN HEMTs do not require doping to attain a high concentration of electrons in the channel. Instead, carriers are created by the polarization mismatch between the GaN and AlGaN barrier layers [43] and by piezoelectric and spontaneous polarization. The 2DEG value for a GaN-based HEMT is significantly greater than that of indium phosphide (InP) or GaAs-based heterostructures [44,45,46], as it is influenced by the physical features of the respective materials [47]. The conduction band of the GaN channel layer is also lower than the energy level of the AlGaNs barrier layer, shifting the balance of the electron transfer toward the channel layer from the barrier layer and confining the transferred electrons to the 2DEG layer.

## 3. Challenges and Opportunities

### 3.1. Normally-Off Operation

Enhancement mode (E-mode), or normally-off operation, is crucial for modern high-frequency HEMT switching, such as in 5G applications. This function provides a more straightforward transistor control system without a negative power supply and advantageous operating conditions for device safety. However, because 2DEG allows for a large current, HEMT devices constructed with an AlGaN/GaN heterojunction are inherently normally-on devices. Even without gate bias, 2DEG remains present at the AlGaN/GaN heterointerface, resulting in a normally-on operation, also called depletion mode (D-mode) [48]. Currently, normally-off transistors are strongly recommended for power electronic applications [49] to simplify the gate drive arrangement and alleviate safety concerns. In cases of gate driver failure, when the gate bias drops to 0 V, a normally-off HEMT shifts to the off state, preventing circuit burnout. This is clearly safer than leaving a switch in the always-on position. To achieve normally-off HEMT behavior, a positive threshold voltage (V_th_) is essential. Despite being more vulnerable to high leakage current, normally-off transistors can attain a high breakdown voltage (V_BR_) of over 1100 V, a significant improvement considering that reported values are often substantially below the theoretical limit [50].

Over the past 20 years, significant efforts have been made to investigate a possible methodology for developing well-built, normally-off GaN HEMT technology. The original inspirations for the study of normally-off HEMTs were the recessed-gate technique with an ultra-thin AlGaN barrier [51] and the injection of fluorine ions into the AlGaN barrier layer [52]. The recessed-gate technique involves reducing the thickness of the AlGaN barrier layer beneath the gate [53,54] because a positive V_th_ and 2DEG depletion are typically associated with a specific AlGaN thickness, at which the Fermi level at the interface becomes lower than the AlGaN minimum conduction band. The reduction of the AlGaN layer thickness through the gate recess leads to a lower polarization-induced 2DEG density in the commonly used AlGaN/GaN heterostructure for HEMTs. With deep enough gate-recess etching, V_th_ may approach a positive value, resulting in E-mode HEMTs [55].

Alternatively, fluorine gate HEMTs use either ion or plasma implantation to insert negatively charged fluorine ions under the gate [56,57]. The negative charge then depletes the 2DEG, reducing channel mobility and electron density. A decrease in the gate leakage current is then expected due to the negative fixed charges of the fluorine ions. This procedure has been further developed by adding a dielectric layer under the gate in the recessed area to decrease leakage current and eventually move V_th_ to the positive side [58]. The device’s V_th_ may be regulated more accurately by adjusting the dielectric layer thickness. However, this technique has several drawbacks, mainly regarding plasma etch process repeatability at a nano-metric level and V_th_ instabilities with a rising operating temperature [59].

Off-state operation can also be achieved by combining a normally-on HEMT in a “cascode” arrangement with a low-voltage, E-mode Si metal-oxide-semiconductor field-effect transistor (MOSFET) [60]. These two devices are coupled such that the MOSFETs output drain-source voltage (V_ds_) directly influences the HEMTs input gate-source voltage (V_gs_). The operating premise of the cascode method is as follows: Immediately after the Si MOSFET is switched on, the normally-off GaN HEMT is switched on. Since the GaN HEMT and Si MOSFET are coupled in series, any voltage provided to the drain terminal causes current flow across both devices. The drain terminal of a GaN HEMT is negatively biased when the Si MOSFET is switched off [61]. However, while the cascode arrangement may be powered by standard MOSFET drivers, it has significant disadvantages. Connecting two devices in series, for example, increases package complexity [62] and introduces parasitic inductances, which influence cascode switching functioning and restrict high-temperature capabilities [63].

Recent technological development has focused on two critical device structures for off-state operation: recessed-gate hybrid metal-insulator-semiconductor HEMTs (MISHEMTs) and p-type GaN (p-GaN) gate HEMTs. In recessed-gate hybrid MISHEMTs, the AlGaN layer is separated from the device by plasma etching at the gate region, and the recessed GaN region is passivated [64]. The resulting device is a hybrid transistor linking two low-resistance access zones to the recessed metal-insulator-semiconductor (MIS) channel due to the presence of the 2DEG. Nevertheless, despite promising achievements, GaN-based, recessed-gate hybrid MISHEMTs suffer from V_th_ instability caused by charge trapping inside the gate insulator [65,66]. The impacts of the instability are two-fold: a positive V_th_ shift degrades the device’s on-resistance since it requires greater bias for the same current value, while a negative threshold voltage shift (positive charge trapping) can cause the loss of normally-off characteristics [67]. Thus, the impact of charge trapping at the gate dielectric is a significant concern for recessed-gate MISHEMTs [68]. As normally-off GaN HEMTs are still novel, this method has not matured enough to market and remains a GaN community research and development (R&D) project.

On the other hand, p-GaN gate design has long been considered the most advanced structure for producing normally-off GaN HEMTs [69] due to its stable threshold voltage and high reliability [70]. This implementation uses p-type GaN or AlGaN with ample acceptor doping on top of the AlGaN [71]. The fundamental structure of p-GaN HEMTs consists of a p-GaN cap layer, metal contact, an AlGaN barrier layer, an undoped GaN layer, and a substrate layer, as illustrated in Figure 2a. By adding a p-GaN layer on top of the AlGaN/GaN heterostructure, the AlGaN conduction band is lifted above the Fermi level by an amount of energy similar to the GaN band gap (3.4 eV), causing 2DEG depletion. To achieve 0 V gate bias, the depletion zone for the given p-type doping extends throughout the GaN channel layer, interrupting the 2DEG at the gate position. Consequently, the GaN HEMT switches from normally-on to normally-off mode. The 2DEG transistor channel is then re-established using positive gate bias, resulting in on-state conditions for the transistor [72]. Figure 2b shows the diagrams of the studied HEMT with a p-GaN structure and its energy band gap.

Figure 3 shows the simulated electric field for a typical p-GaN HEMT. The active devices that comprise power-switching systems must be normally-off for a higher level of inherent safety (i.e., a positive threshold voltage, V_th_). However, adding a p-GaN layer to an AlGaN/GaN heterostructure is usually insufficient to achieve normally-off behavior. Instead, various factors must be considered, including heterostructure characteristics, thermal annealing, gate contact, p-GaN etching, and doping [73].

As previously mentioned, the selection of metal gates in normally-off HEMTs with p-GaN gates can also significantly influence a device’s performance and long-term reliability [74]. Overall, there are two possible types of gate contacts in p-GaN gate HEMT technology: ohmic and Schottky. A Schottky contact is a rectifying contact between a metal and a lightly doped semiconductor. By modulating the metal with a specific gate voltage, the charge density of the heterostructure and the drain current can be conveniently adjusted [75]. The drain current must be switched off by setting the Schottky gate HEMT to a reversed bias, as most GaN-based HEMTs are “usually-on” devices due to the intrinsic characteristics of the AlGaN/GaN heterostructure [76]. The bending of the bands at the interface then creates a Schottky barrier. The Fermi levels in the two materials should be matched at thermal equilibrium whenever a metal or superconductor comes into close contact with a semiconductor. The metal/p-GaN Schottky barrier height is usually proportional to the device’s threshold voltage (V_th_). However, an ohmic contact can be formed by heavily reducing the Schottky barrier, enabling current conduction in both directions without rectification. In other words, a HEMT with either a Schottky or ohmic gate can be made by strategically structuring the Schottky barrier.

Overall, it is generally agreed that a Schottky contact is more practical for a p-GaN HEMT than the ohmic contact since the latter induces a relatively large gate leakage current and a lower threshold voltage [77]. Figure 4 depicts Schottky and ohmic contacts for a p-Gan HEMT [78]. The p-GaN layer thickness, acceptor concentration, and gate metal work function (*M*) are the primary design factors regulating the geometry of the Schottky barrier. These factors impact threshold voltage, breakdown voltage, and transconductance (g_m_) [79]. In p-GaN gate HEMTs, a good Schottky barrier assures the lack of substantial current injection at the gate side, resulting in a decreased power consumption [80]. Hence, the Schottky gate solution on p-GaN is preferred over the ohmic gate solution.

In addition to establishing off-state characteristics, optimizing the performance of a p-GaN HEMT is also a priority. This section considers factors that significantly influence threshold voltage values, including AlGaN barrier thickness and p-GaN doping concentration. Additionally, since both the on-state resistance and the threshold voltage vary depending on the electron concentration (n_e_), there is a known trade-off. Table 2 shows compilations of normally-off p-GaN gate HEMTs from recent literature.

The highest threshold voltage and drain current were achieved by Panda et al. [81], with a much thicker p-GaN being implemented. Due to thermal and processing compatibility, stacked Ti/Au or Ni/Au metal gate contacts have generally been preferred for p-GaN HEMT. Chang et al. [85] illustrated the advantages of these gate contacts, achieving a positive threshold voltage shift of 2.2 V in p-GaN HEMTs with Ni/Au gates. Efthymiou et al. [95] also found that a Schottky contact at the p-GaN gate could reduce the gate current by several orders of magnitude and result in a higher gate bias and gate turn-on than with an ohmic contact. When a potential drop is detected across the p-GaN cap layer depletion area, a larger bias voltage with a Schottky gate contact is required to minimize the potential barrier at the p-GaN/AlGaN interface. However, this contradicts the recommendation by Tsai et al. [96], who preferred a hybrid Schottky–ohmic gate contact. A lower gate turn-on voltage is associated with a larger gate metal work function. Increasing acceptor doping does not influence the device’s threshold voltage but alters the gate turn-on voltage at high p-GaN doping levels [95]. As p-GaN doping increases, a device’s threshold voltage at first rises, but with a further increase, it begins to drop. Tight electrical connectivity between the p-GaN layer and gate metal is established through hole tunneling at the metal/p-GaN interface. Introducing a different gate metal cannot appreciably modify the threshold voltage at high doping levels.

However, compared to previous findings, Chiu et al. [97] achieved the best electrical properties in a p-GaN HEMT design by implementing the deposition of an Al_2_O_3_/AlN gate insulator layer through the atomic layer decomposition (ALD) process. Figure 5 illustrates the authors’ device structure, which achieved a very high threshold voltage (3 V) and saturation drain current (around 363 mA mm^−1^). The turn-on voltage in this study was also higher than 20 V, while the gate leakage current was reduced. Overall, these electrical properties are better than those obtained in many other studies [98,99,100]. Hence, including an Al_2_O_3_/AlN layer created via ALD helped build a good interface between p-GaN and AlN, increasing the device’s off-state V_BR_ in the MIS gate.

Furthermore, the AlGaN barrier thickness plays a significant role in determining the threshold voltages of p-GaN gate devices. A GaN-based HEMT is typically constructed with a single AlGaN barrier layer with a thickness range of 10–15 nm. However, none of the researchers had considered Al composition within the AlGaN barriers until Wu et al. [101] developed HEMTs with a double barrier layer with consideration for the concentration of Al. Table 3 shows several recent studies on multiple-barrier GaN-based HEMTs. The gate dielectric is also critical for GaN-based MISHEMTs [102]. Although many studies have focused on the gate dielectric material, only a few have considered the importance of length and recessed depth. As seen in Figure 6, Xia et al. proposed a triple barrier layer [103] with a variation of Al content between 15% and 25%, showing superior DC characteristics. As a result, the particular size of the gate dielectric in the device manufacturing process must be further investigated.

Several methods of fabricating p-GaN HEMTs have been developed. A stacked-gate self-aligned patterning technique is commonly used to etch the stacked metal gate and the p-GaN in the same sequence [106]. Essential procedures include the selective etching of overgrown p-GaN layers and the repair process. A high etching selectivity ratio is typically required for the p-GaN HEMT etching technique, as both over- and under-etching negatively impact device performance [107]. Under-etching causes the residual p-GaN layer to deplete the 2DEG to some extent, while over-etching causes the 2DEG density to drop due to a thinner AlGaN barrier layer [87]. Both cases eventually impair conduction. Hence, different approaches have been explored to resolve this concern. For instance, Niu et al. [91] assessed various repair methods and recovered electrical properties by up to 93%. Furthermore, to account for potential surface damage and reduced amplifying effectiveness [108], a solution of backside dry etching was proposed [109], leading to a maximum increase in saturation current density and g_m_ by 21.1% and 25%, respectively. Surface damage may also be minimized by using a selective inductively plasma process (ICP) with a mixture of boron trichloride (BCI_3_) and sulfur hexafluoride gas (SF_6_) [110,111]. A recent study by Osipov et al. [112] suggests that stress may also alter 2DEG concentration and thus the electrical properties of AlGaN/GaN HEMTs, because of the piezoelectric nature of GaN. This theory was further proven in another study [113], demonstrating that dielectrics liner stress may cause many piezoelectric charges within the heterostructure underneath the gate metal. Hence, strain engineering is considered an effective method to improve threshold voltage with a scaled gate length.

In sum, devices with normally-off characteristics are highly recommended for power switch applications to assure fail-safe operation. A reliable normally-off HEMT technology is essential for the long-term widespread use of GaN transistors. Due to the favorable trade-off between reliability and cost, the p-GaN gate HEMT is currently the only viable solution [114]. However, various issues, such as threshold voltage instability [115] and increased off-state leakage current persist due to the on-state gate bias [116]. However, the charge-transferring effect [117] of the charge control model may explain the threshold voltage instability. Additionally, high positive threshold voltages are difficult to attain due to the trade-off between the threshold voltage and sheet resistance in the channel [118].

### 3.2. Self-Heating Issues

With the modernization of semiconductor technologies, designers have continually increased the power density of power devices, leading to increased channel temperatures and decreased drain currents (I_ds_). For GaN-based HEMTs, a high drain bias (V_D_) is used for high-power and high-frequency applications, producing a strong lateral electric field from the drain electrode side at the gate edge. As a result, the local lattice’s temperature rises, a result that is known as the self-heating effect. In practice, extreme overheating eventually reduces the lifetime of GaN devices or causes irreparable damage [119], significantly impacting long-term use [103]. Therefore, commercial GaN HEMTs are currently restricted to 2–4 W mm^−1^ output power, compared to the proven 40 W mm^−1^ power output as power amplifiers. Lowering the structural temperature would therefore enhance devices’ power efficiency and reliability in the long run [120,121,122]. Amar et al. [123] shared the same concern, believing that HEMT technology failures are primarily linked to operational temperatures exceeding critical levels due to component self-heating. Self-heating may also cause other issues, including gate burying, connection chip-package damage, electron mobility degradation, and current decrease [124]. Thus, thermal management is critical at the design stage [125,126,127,128] to limit performance degradation and increase reliability [129,130].

Improving the thermal design of AlGaN/GaN HEMTs requires precisely estimating the underlying thermal transport mechanisms. When multi-layer architectures with a low thermal conductivity are used in the HEMT structure, they impede heat dissipation from the junction to the substrate [131], increasing the relevance of the GaN layer for effective heat removal. Since heat is created solely around the gate, and the gate length is less than a micrometer, proper temperature monitoring necessitates spatial resolution on a scale of 1 μm or less. Practically, this procedure is possible with micro-Raman spectroscopy and thermoreflectance thermal imaging [132], which allow designers to quantify channel temperatures and map a device’s temperature distribution with a spatial resolution in range. Lundh et al. [133] and Chatterjee et al. [134] used the same method, measuring the lateral and vertical steady-state operating temperatures of AlGaN/GaN HEMTs. Their outcomes revealed that channel temperature could not be calculated exclusively by continuous scale heat transfer principles due to the interaction of heat concentration and subcontinuum thermal transport. It has since been proposed that nanowire-channel HEMTs reduce the temperature dependence and overall threshold voltage for better temperature stability [135]. These findings may be used to assess self-heating effects in HEMTs and as a reference for further improvement.

From a structural perspective, thermal improvement is typically influenced by the substrate materials on which HEMT devices operate. For GaN-based HEMTs, epitaxial layers are commonly grown on a foreign substrate, such as sapphire, silicon (Si), silicon carbide (SiC), or diamond [136]; Figure 7 illustrates the thermal analysis for some of these materials [137]. These dissipative substrates help suppress thermal mismatch while improving thermal stability [138]. It is also determined that a highly resistive substrate may enhance breakdown robustness, but there is always a trade-off between threshold voltage stability and material cost [139].

The most notable substrate materials used to determine the effect of dislocation on thermal behavior are Si and sapphire, preferred due to their low costs [140]. One of the most noteworthy accomplishments using Si was made by Xing et al. [141], who achieved an f_T_ of 250 GHz, the highest for GaN-based HEMTs, on Si with deeply scaled gates. Furthermore, they achieved a 25% increase in output current and a 40% reduction in heat. GaN-on-Si structures are also promising for vertical HEMTs because they could reduce switching loss (E_sw_), which accounts for significant power loss and device temperature, especially under high-frequencies. Compared to lateral structures, vertical devices have much simpler thermal management [142] but are significantly more difficult to demonstrate on foreign substrates than on native GaN substrates [143]. Therefore, GaN-on-Si HEMTs have been considered the overall best-in-class power semiconductors [144] despite severe limitations due to losses associated with output capacitance [145].

However, the poor thermal conductivity of Si and sapphire restricts heat dissipation during HEMT operation, which may affect electrical performance and reliability [146]. For this reason, replacement substrates with better thermal conductivity, such as GaN or SiC [147], are commonly used. Broad band gaps make these replacement substrates better than Si for very high-temperature operations (up to 600 °C versus 200 °C) [132]. Moreover, their high-power densities can be successfully dissipated at realistic drain efficiencies, avoiding the severe channel temperatures generated by other substrate technologies due to self-heating. Figure 8 illustrates a HEMT structure grown on a heat-dissipating SiC substrate, with the simulated thermal modeling shown in Figure 9. Another benefit of a SiC substrate is that it has a reduced lattice misfit of 3% for GaN, compared to 17% for Si. Hence, devices using GaN and SiC substrates are predicted to function favorably in high-temperature conditions due to their better material characteristics [148].

Less favorably, commercial GaN and SiC bulk substrates have been relatively challenging and expensive to acquire [149]. However, Huang et al. [148] addressed pricing concerns by proposing a low-resistivity SiC (LRSiC) substrate. This proposal has several benefits over Si HEMT, including a larger output current, a higher off-state, a higher vertical breakdown voltage, and a lower dynamic specific on-resistance ratio, which are vital for thermal performance. The LRSiC substrate is also three times less expensive than a standard SiC substrate, as shown in Table 4. Thus, it may be an excellent solution to the heat and cost problems associated with power devices.

AlN can also be implemented to improve heat dissipation in the HEMT design. For example, Cheng et al. [150] investigated AINs inherent thermal conductivity by growing a thick film of AlN on sapphire substrates, improving heat dissipation. This research agrees with a study by Chang et al. [151], who operated GaN-based HEMTs on an AlN substrate or a Cu film, improving electrical and RF performances such as the g_m_, the drain current, the f_T_, and the maximum oscillation frequency, as seen in Table 5. The reduction of self-heating helps increase carrier mobility beneath the gate and reduce sheet resistance at the access region, promoting electrical improvement [152,153].

Diamond is also a suitable substrate for further reducing the self-heating effect because of its high thermal conductivity (up to 2000 W m^−1^ K^−1^) [154]. Integrating AlGaN/GaN thin-film transistors onto diamond substrates improves heat dissipation and device performance and reliability. For instance, Gerrer et al. [155] have tested this approach, which allowed for more effective heat dissipation, improving performance and reliability with a significant GaN-on-diamond output power of 14.4 W at a P_out_ of 8.0 W mm^−1^. The relationship between the geometric parameters of GaN-on-diamond substrates and junction temperature was observed, particularly in relation to diamond thickness. The alteration in thickness correlates to changes in the distance between the diamond and the heat source edges and, thus, to changes in the junction temperature. Hence, as the thickness of the diamond substrate increases, the temperature (*T*) proportionally decreases [156]. However, the epitaxial development of a diamond substrate is typically more complicated and costly than a SiC substrate [155,157].

A better thermal design is also possible through the construction of HEMTs with Cu-filled structures. Jang et al. applied two different Cu-filled thermal designs [137] under the active portion of the basic GaN-on-SiC (BGS) HEMT, as illustrated in Figure 10. The 2DEG channel’s lateral and vertical lattice temperatures were addressed during device operation, followed by a transient thermal analysis. Figure 10a shows a BGS device, whereas Figure 10b,c illustrates the two thermal structures in the SiC substrate. SiC substrates beneath the active area were etched away, forming Cu trenches or vias. This thermal design benefits from the control of steady-state thermal parameters, such as the vertical lattice temperature, the lateral lattice temperature inside the 2DEG channel, and the heat production rate as power density increases. Overall, Cu-filled thermal structures have lower maximum junction temperatures and attaching thermal structures to GaN HEMTs reduces the time to achieve the maximum lattice temperature. Thus, implementing Cu-filled thermal vias (CTV) improves heat regulation.

Alternatively, thermal improvement is possible with the modification of AlGaN barrier layers. Since GaN and AlGaN layers are much thinner than the substrate layers, their effects on channel temperature should be less substantial. However, the AlGaN layer still controls substrate heat dissipation capability due to the differing thermal conductivities of AlGaN and GaN [158]. An increased room temperature thermal resistance often instigates higher device self-heating and broader temperature gradient layers due to the reduced thermal conductivity of the AlGaN. Wang et al. [159] proposed a viable solution by introducing a back-barrier (BB) layer to the buffer layer, thus limiting the impact of the doped acceptor between the channel and buffer layers. Apart from the thermal improvement, the withstand voltage was also enhanced, which, in turn, decreased the current collapse effect.

For another thermal solution, Chvála et al. [160] proposed a multi-finger power HEMT structure with thermal crosstalk among several individual gate fingers. This structure may help raise structural temperature and reduce power density with compact multi-finger layouts. They considered various thermal bottlenecks in GaN-based HEMTs, including a lower thermal conductance of transition layers, heat transport across interfaces, and thermal conductivity from phonon-scattering processes. Additionally, a commercially available engineered substrate, Qromis Substrate Technology (QST), has already been proven to mitigate the impact of low heat dissipation [161]. The overall thermal resistance of QST substrate is lower than that of Si substrate due to its higher thermal conductivity, which may lessen the influence of heat on a device. Micro-trench structures packed with Cu can also be modeled to offer a heat escape path from any hot region, leading to considerable improvements in electrical performance [162]. As shown in Figure 11, heat generation can be firmly focused within the channel on the drain side of the gate, hence dramatically lowering temperatures in these hot areas.

In short, self-heating is a critical concern in HEMTs due to the possibility of locally reaching a high power density and a non-uniform thermal dissipation. This concern is also supported by the fact that many of these devices’ features, including electron mobility, the saturation rate, and the thermal conductivity, are temperature-dependent [163]. Thus, thermal behavior significantly influences a HEMTs long-term reliability [164,165], as shown by the possible gate burying, deterioration of the feed metal interconnection, and degradation of the Schottky contact, which all eventually impact the failure rate [166,167,168]. Given the availability of various HEMT substrates to improve thermal behavior, a significant trade-off exists between performance and manufacturing costs.

### 3.3. Current Collapse, Peak Electric Field Distribution, and Gate Leakage

Another critical challenge for a GaN HEMT is current collapse, or on-state resistance (R_DS,ON_) dispersion [169]. The fundamental source of this issue is the formation of a virtual gate between gate and drain terminals. When the gate and drain voltages are adjusted rapidly, slow current transients can occur, often referred to as gate lag and drain lag [170]. This issue manifests as an increase in the dynamic on-state resistance in switching devices [171] and significantly affects a device’s long-term reliability.

Furthermore, there are also issues with the high peak electric field, which occurs at the gate edge of the drain side during operation under high bias circumstances [172]. This high electric field may facilitate charge trapping between the passivation layer and III-nitrides interface. Electrons may also become stuck in free surface states under a strong electric field, triggering virtual gating and current collapse [173]. Owing to smaller gate–drain spacing, devices undergo significantly higher current collapse when scaled down for high-speed operation, amplifying the virtual gating effect of surface traps. Moreover, controlling the electric field distribution between the gate and drain is critical for obtaining a linearly scaled breakdown voltage per channel length. Scaling high-power GaN-based HEMTs to achieve low on-resistance and gate charge (Q_g_) is thus still a challenge for high-power and high-speed operation. Hence, the peak strength of the electric field at the gate edge must be reduced to achieve a high breakdown voltage [174].

Likewise, the gate leakage current is an essential parameter for GaN HEMTs and is directly linked to device performance and reliability. Forward gate leakage current restricts the gate voltage swing and results in drive losses, while the reverse may result in off-state power consumption and a reduction of V_BR_ [175]. Excessive gate leakage currents are not permitted, as they may lead to unwanted power consumption. Therefore, setting the Schottky gate to a reversed bias can help evade potential power loss by switching off the drain current. For this reason, research studies on gate leakage mechanisms are commonly linked to peak electric field distribution and current collapse concerns [176]. The supply of active electron traps between the gate and drain decreases dramatically with the peak electric field, resulting in a lower current collapse and knee walkout [177]. Large band gaps and significant band offsets for gate insulators are thus required to suppress gate leakage current, even at forward bias.

Some viable solutions to address these concerns include FP implementation, surface passivation, and gate structure variations. FP refers to an extension of the gate deposited onto the passivation layer toward the drain side, where the electric field at the AlGaN surface decreases. As shown in Figure 12a, the metallization layer sits on top of the passivation layer of HEMTs and prevents the current collapse effect by reducing the peak electric field near the gate’s drain edge [178]. In theory, the profile of the electric field distribution improves as FP successfully broadens the depletion region with multiple peaks that may substitute for a single peak, resulting in a more uniform electric field distribution [179]. FP implementation also helps reduce reverse leakage current. By providing an extra surface for field line termination and thus dispersing the electric field over a longer gate-to-drain interval, FP can reduce the maximum electric field and lessen electrical field congestion at the drain side of the gate edge.

To date, various architectures of HEMT have used FP. For instance, Zhang et al. [180] discovered that FP technology might give lateral power devices a novel charge-balancing effect. Wong et al. [173] created a GaN HEMT with an innovative asymmetric slant FP, achieving a high breakdown voltage of 146 V with the aim of increasing the breakdown voltage without increasing the device size. This outcome was consistent with a study by Chen et al. [181], which found that the potential dispersion near the drain edge grew as the source voltage increased, caused primarily by the increasing electric field between the gate and drain areas. There was also a rise in potential near the standard FP edge, resulting in an extremely high electric field of 4.8 MV cm^−1^. Kabemura et al. also investigated this topic [182] and saw an enhancement in breakdown voltage when using short- and moderate-length FPs on HEMTs. Table 6 reports multiple recent findings on the characteristics of HEMTs using FP and highlights the importance of optimizing devices’ geometrical parameters.

As shown in Table 6, few FP HEMTs have employed a GaN cap layer to help suppress self-heating effects and current collapse. This layer also shields the AlGaN surface from oxidation, offers an extra barrier at the Schottky contact, and decreases leakage current [195]. The concept could be further enhanced with a high-resistivity GaN cap layer, which can improve the electric field distribution, current collapse, and breakdown capability, resulting in a high V_BR_ of 1020 V [196]. Nirmal et al. [184] investigated this theory further by adding an AlN layer between the SiN and GaN layers, as shown in Figure 13a, resulting in a 6.26% increase in drain current compared to the conventional design. Breakdown voltage was also 14% higher, while the current collapse was reduced by 10%. These improvements were caused by the AlN cap layer, which can accommodate more heat than the GaN cap layer due to its better thermal conductivity of 2 W cm^−1^ K^−1^. Thus, a sandwiched AlN layer helps reduce lattice mismatch and trap charges at the SiN/AlN interface, ultimately improving the proposed HEMT design.

FP technology has been continually modified to improve performance. Wong et al. [173] recently developed an exceptional SiN slant FP on AlGaN/GaN HEMTs by employing the surface tension properties of hydrogen silsesquioxane (HSQ) on a pre-patterned plasma-enhanced chemical vapor deposition (PECVD) SiN dielectric. With f_T_/f_max_ = 41/100 GHz, the resulting HEMT with a tuned slant FP displayed a very low dynamic-specific on-resistance and a solid high-frequency performance. Augustine Fletcher et al. [179] achieved a similar result using a discrete FP with part of the lateral plate removed, as shown in Figure 14. With a high breakdown voltage of 330 V, compared to 298 V in a conventional design, the discrete FP reduced the maximum electric field between the gate and drain regions. Furthermore, the FP gate HEMTs leakage current was around ten times lower than that of the non-FP design. This lower current may be attributed to the FPs smooth electric field distribution, which effectively lowers the inverse piezoelectric and electron tapping effects in the AlGaN barrier layer. The improvement can also be attributed to fewer defects generated via gate leakage.

Soni et al. performed further research on FPs [197] by comparing three designs: a drain-connected lateral FP, a drain-connected vertical FP, and a dual-FP structure. A significant breakdown voltage roll-off was observed after increasing FP length in a lateral design due to a change in the peak electric field from the drain edge to the gate edge. This was followed by an increase in the peak electric field at the gate edge. In a drain-connected vertical design, the breakdown voltage is restricted by the buffer thickness, resulting in a breakdown voltage roll-off as the FP thickness increases. However, both concerns are addressed by the dual-FP structure, which allows the electric field to be shared over the gate and drain side, especially in scaled designs. This enables the scaling of HEMTs with a dual-FP architecture, improving the on-state performance without sacrificing the breakdown performance.

Xia et al. [198] investigated the potential of micro-FP technology. They found that the suggested technique may offer a charge balancing effect for HEMTs with better performance than a lateral structure. This outcome is due to the impact on the potential distribution, resulting in an expanded electric field distribution between the gate and drain and a peak electric field concentration at the micro-drain field plate (D-FP), the gate field plate (G-FP), and the source field plate (S-FP) edge. Figure 15 shows the schematic diagram of HEMT with a micro-FP structure.

FP remains in contact with the passivation layer made of nitride or oxide, preventing electron leakage with high-density shallow surface traps [199]. A passivation layer such as Si_3_N_4_ can be formed with the metal-organic chemical vapor deposition (MOCVD) method, which helps lessen the influence of surface states that restrict saturation current and the device’s breakdown voltage [200]. Hence, using a passivation layer can improve the saturation current, the breakdown voltage, and the noise level [201]. However, several elements of a passivation layer’s action mechanism, such as layer thickness, are still being debated [202].

When designing HEMTs, it is also essential to include a gate insulator layer between the AlGaN barrier layer and the gate metal to reduce gate leakage [203]. For gate insulator applications, various dielectric materials have been investigated, including hafnium dioxide (HfO_2_), silicon dioxide (SiO_2_), aluminum oxide (Al_2_O_3_), silicon nitride (SiN*_x_*), and zirconium dioxide (ZrO_2_). High-k dielectrics, in particular, promise particularly beneficial channel controllability for low off-state leakage currents, high on-to-off ratios, and low SS, suggesting improved power efficiency in device applications [23]. Table 7 shows dielectric characteristics for commonly used high-k materials [204]. However, although the threshold voltage may be raised, many gate-related adjustments result in undesirable side effects such as excessive gate leakage and low gate swing. High-quality gate dielectrics are therefore required to minimize gate leakage and retain the inherent high mobility of 2DEG, particularly in the recess gate structure, which often oversees scatterings from a poor dielectric/GaN surface, resulting in decreased gate reliability. Further, an additional gate dielectric layer usually results in more complicated material interfaces, and the interface quality substantially influences the device’s electrical properties [205]. Hence, the interface quality of HEMT devices warrants further study.

Currently, HfO_2_ is the most extensively used high-k gate insulator, particularly in the Si complementary metal-oxide semiconductor (CMOS) industry, due to its high-k value and large band gap (5.8 eV). However, using such a dielectric layer typically increases the complexity of the new interface; HfO_2_ suffers from extreme oxygen transparency, which introduces unfavorable Ga–O bonds in GaN-based devices and degrades the condition of the HfO_2_/GaN interface [206]. While HfO_2_ HEMTs can attain more efficient electrostatic control, they suffer from excessive leakage current owing to an inadequate barrier height, which degrades device performance through gate leakage. This drawback was noted by Huang et al. [207], who recommend devices with SiN_x_ gate dielectric over those with HfO_2_ due to a better electrical stability and a low threshold voltage drift resulting from a lower interface trap density. Therefore, the superior electrical stability of the MIS-HEMTs with SiN*_x_* gate dielectrics can be attributed to their greater interface quality. However, this information might not be accurate across all applications, such as in the space industry, where the impact of proton radiation on a HEMT device using a dielectric layer must also be considered. As per a study by Lee et al. [208], proton irradiation induces negative charges in gate dielectric layers, which can degrade certain performances of MISHEMTs, such as threshold voltage shift and the reduction of drain current. This investigation indicated that the Al_2_O_3_ dielectric layer is considerably more suitable than SiN_x_ as a gate insulator for AlGaN/GaN MISHEMTs in space applications since the increased induced charge density in the MISHEMT is not severe, resulting in less degradation of electrical properties. However, it is still feasible to enhance the dielectric behavior of HfO_2_ by incorporating Si into the dielectric layer, improving the breakdown strength and interface properties. A study by Li et al. [209] found that including Si in HfO_2_ reduced the fixed oxide traps and interface trap density within the dielectric, thereby boosting the breakdown properties of the dielectric.

To further advance high-k metal gate technology, high-mobility substrates for CMOS technologies, such as III-V compound semiconductor materials, have also been investigated. The direct deposition of high-k dielectric can reduce the burden of finding a stable oxide such as SiO_2_. However, due to the intrinsic features of III-V surfaces and their oxidation chemistry, fabricating the high-k/III-V material interface is very challenging and typically results in a high interface state density, leading to a higher concentration of interface states with Fermi-level pinning. Overall, interface properties appear to depend on the deposition technique, a combination of deposition parameters, the substrate surface orientation, pre-deposition surface treatments, and the subsequent annealing temperatures based on the electrical performance [210].

Inserting a high-permittivity passivation layer or a material with a high dielectric constant (k) to boost breakdown behavior should also be considered, as this directly influences the smoothness of the electric field profiles between the gate and the drain. As the electric field at the drain edge of the gate is lowered, the breakdown voltage rises with ε_r_. The breakdown voltage is also enhanced in the high ε_r_ area when the gate voltage is more negative since the buffer leakage current is likewise lowered [182]. There is a direct correlation between the band gap and the permittivity for materials widely used as passivation layers, as seen in Figure 16, which emphasizes their trade-offs [211]. Given the massive difference in permittivity between high-*k* dielectrics and AlGaN, a high-k film should be able to transmit or extract electric flux more effectively from the semiconductor surface. Multiple studies have also demonstrated this impact by implementing different passivation layer materials, such as MgO, SiO_2_, ZnO, and Si_3_N_4_, with varying *k*. It is noteworthy that the drain current typically increases with high-k passivation layers. Furthermore, the surface effects are reduced, boosting the channel carriers and increasing the drain current [212]. This outcome agrees with the usage of high-k dielectric material as a passivation layer, ultimately reducing the dynamic specific on-resistance or current collapse while improving breakdown voltage [213].

Implementing multiple passivation layers in a single HEMT device may also improve performance. In one study [214], the breakdown voltage for a double passivation layer structure was enhanced significantly against a single passivation of SiN due to the weakened electric field around the drain edge of the gate. The stack passivation layer of Al/SiN also minimizes damage at the AlGaN surface. A similar improvement was observed in another study [215] with an Al/SiN stack layer, in which the gate leakage was reduced by several orders of magnitude, effectively suppressing moderate current collapse and improving the breakdown voltage by 32.8%. Murugapandiyan et al. [216] studied a dual SiN/AlN passivation HEMT with a self-heating model and showed a 60% increase in drain current density and a 63% increase in g_m_; thus, they found that the model was reliable and stable for an extensive range of operations.

A charged passivation layer (CPL) has also been considered for a GaN HEMT structure to enable higher modulation of the electric field distribution along the channel layer, thus improving the homogeneity of the electric field along the entire channel [217]. In general, CPL HEMTs outperform traditional structures in breakdown voltage, frequency performance, and specific on-resistance. In addition, a study reported that a graphene layer (GL) could be mounted above the SiN passivation layer of HEMT, improving devices’ thermal management [218]. Due to the excellent hydrophobic properties, trapping effects are efficiently prevented, particularly those that are water-related. This implementation allows for a thinner SiN layer, reducing fringing capacitance without compromising water-related current collapse effects. These findings emphasize the importance of the GL in increasing the SiN passivation layer’s moisture resistance while maintaining the AlGaN/GaN MISHEMT’s electrical properties.

Several new ideas concerning barrier layer variations have been introduced to address current collapse and E-field distribution concerns. A novel design of enhancement-mode GaN HEMT with a thick GaN buffer and a step-etched GaN structure (SGB) has been explored [219], resulting in improved forward output characteristics. As shown in Figure 17, a thin GaN buffer without a step structure (TGB) and a conventional GaN buffer structure (CGB) were developed for comparison. As the GAN buffer’s thickness increased, the breakdown voltage and maximum current drive capacity increased only marginally, as detailed in Figure 18. A recent proposal implemented an ultra-thin barrier (UTB) and a local charge compensation trench (LCCT) [220]. Deeper and longer LCCTs produce more negative charges, resulting in a high E-field redistribution capacity. Hence, any potential lattice damage in the barrier might be avoided. This topology modulates the 2DEG concentration to smooth the reverse E-field by injecting additional negative charges.

Current collapse in GaN HEMTs can also be addressed through notch formation between the gate and drain. A notch in the AlGaN barrier layer may help reduce the 2DEG concentration inside the channel while suppressing the peak electric field alongside the gate electrode. Figure 19 illustrates a basic HEMT with a single notch structure. Zou et al. [221] investigated the impact of the dimensions and numbers of notch structures by covering six models with varying notch designs. Compared to the conventional AlGaN/GaN HEMT, double-notch HEMT showed the most significant DC and RF performances, including increases of 30% in gate voltage swing, 42.2% in breakdown voltage, and 9% in f_T_, in addition to strong suppression of the current collapse.

It is also essential to address the challenges outlined above through structural variation of the gate. The focus is on efficiently distributing the electric field while effectively managing current collapse and other electrical properties, such as breakdown voltage and g_m_. A significant electrical improvement has been observed when the gate structure has been changed from an FP to a gamma gate [222]. A similar outcome was recorded for a slanted tri-gate design [223], which efficiently distributed the electric field and significantly enhanced the breakdown voltage. This gamma gate structure can be engineered through lithography by adjusting the width of the tri-gate nanowires. Accordingly, the impact of the gate length has been addressed in two commercial Gan/AlGaN HEMT devices with different gate lengths, as listed in Table 8. Both models ultimately share the same voltage breakdown behavior and drain current; therefore, no differences are expected regarding gate leakage or current collapse [224].

Increasing the number of gate contacts may also reduce current collapse and address high peak E-field issues. For instance, a dual-metal-gate (DMG) construction is superior to a typical single-metal-gate (SMG) structure for achieving the channel’s appropriate electric field distribution. Accordingly, the E-field with a DMG structure is better distributed due to its improved ability to suppress current collapse while boosting overall electrical properties [225]. This information also applies to a tri-gate structure [226] coupled with a hybrid ferroelectric charge trap gate stack. Due to electrostatic control by trapped charges in the charge-trapping layer on the nanowire sidewalls and optimization of the tri-gate form, this structure exhibits a low current collapse and robust electrical characteristics. The hybrid ferroelectric charge trap gate stack also provides a high density of negative charges, resulting in a high positive threshold voltage. However, this result contradicts another study [227] in which a triple material gate (TMG) HEMT provided a lower threshold voltage than SMG and DMG HEMTs. The application of a comb-gate design within AlGaN/GaN HEMT devices has also been proposed, which effectively reduces the off-state leakage current by three orders of magnitude. However, this topology may be insufficient due to the limitation of breakdown voltage, even if the short-channel effect is suppressed. Nonetheless, the quasi-normally-off comb-gate devices are superior to the typical design in regard to switching characteristics and on-state performance, particularly on-state resistance, in the absence of recess operations for device setup [228]. Another study [229] explored the idea of combining gate and ohmic recess. Due to the achievable maximum E-field and electron mobility with a reduced gate channel distance, they were able to attain a low leakage current.

Trench formation is also key to demonstrating high-voltage behavior while addressing some of the challenges associated with HEMT devices. A trench is typically formed between the nucleation and GaN layers, as shown in Figure 20. Zhang et al. [230] presented two trench structures to identify the impact on blocking capability. They found that a flat-bottom rounded trench is the best option for high-voltage vertical GaN power devices, with the lowest possible gate leakage current and the highest breakdown voltage of 500 V. Yang et al. [231] also suggested a novel method of preventing electrons from becoming trapped in the GaN buffer by developing a deep-source metal trench in the GaN HEMT structure. Four device structures were used for comparison: a conventional HEMT, a device with a deep-source contact trench within the mesa area, and double-gate HEMTs with and without the trenches outside the region. Improvement in the current collapse was observed for devices with the source trench within the mesa due to the redistribution of the electric field profile.

To summarize, the challenges associated with current collapse, leakage current, and high peak E-field may be addressed through different structural solutions. To mitigate the impact of ionization on breakdown voltage, for example, it is essential to control the peak electric field. This can be achieved by implementing the correct FP approach. As reported in many studies, the FPs’ purpose is to disperse the electric field profile and lower the electric field peak value, thus minimizing trapping while enhancing the breakdown voltage. A better gate control performance will eventually cause the threshold voltage to become positive and the breakdown voltage to rise due to the smooth electric field distribution. However, the influence of the FP on the electric field is still affected by various factors, including the device’s architecture, thickness, doping concentration, and the k of each layer, which all influence the electric field distribution. Other structural solutions using gate structures, barrier layers, trenches, and notches are also possible.

### 3.4. High Ohmic Contact Resistance

Another challenge associated with HEMT devices is the high level of ohmic contact resistance. In basic terms, the ohmic contacts of HEMT are the device’s access points for connecting to external circuits. Ideally, their resistance should be very low compared to the channel drift region in order to lower the device’s specific on-resistance. Hence, the source and drain ohmic contact resistance (R_c_) should be kept as low as possible for high-power HEMTs [232]. However, the enormous band gap, which naturally favors Schottky connections, makes it difficult to produce excellent ohmic contacts on GaN-based materials. The resulting output power, power efficiency, frequency responsiveness, and noise performance are all known to depend on a low drain ohmic contact resistance. As a result, the work function and thickness of metal layers, the semiconductor doping level, the annealing temperature, the recess depth of the barrier layer, and other parameters require further optimization.

In recent years, different metallization strategies have been employed to achieve low contact resistance. Principally, it would seem that adding Si into the AlGaN barrier would help reduce the ohmic contact resistance. However, when the contacts are annealed at high temperatures to activate the Si dopants, the dopants immediately diffuse away from the contacts, resulting in an increased gate leakage current and charge trapping. Thus, the development of ohmic contacts in AlGaN/GaN heterostructures remains the only solution for ohmic contact resistance issues in modern GaN technology. This is a significant barrier to developing Al-rich AlGaN transistors, as the process of ohmic contact formation becomes significantly more complicated in the presence of an AlGaN barrier and 2DEG [233]. To fully comprehend the development of ohmic connections in heterostructures, one must also fully consider Al concentration and the thickness of the AlGaN barrier layer, which affect features of the 2DEG. The electron affinity in AlGaN transistors decreases when the Al concentration increases, causing massive Schottky barriers at the metal–semiconductor interface. As a result, though shifting to a higher Al composition has numerous advantages, forming an excellent ohmic contact becomes progressively challenging [234].

The combination of multiple ohmic contact materials of HEMT devices is considered the most significant influence on the contact resistance, with multi-layer materials typically applied. The traditional ohmic contact in GaN HEMTs is formed with a Ti/Al/Ni/Au metal stack [235,236,237,238] due to its ease of evaporation and superior electrical properties. Conducting intermetallic titanium aluminide (Ti–Al) is thought to aid electron transport mechanisms, resulting in a low ohmic contact resistance. However, since a HEMT with Au-based ohmic contact is typically incompatible with the latest CMOS technology, recent research has shifted toward Au-free ohmic contacts [239]. An example of this alteration is the combinations of Ti/Al and Ti/Al/Ti/W metal schemes, which lead to superior electrical performance [240], improving maximum drain current by 40.7% compared to conventional structures. It has also been discovered that adding a Ti/W cap layer on top of the Ti/Al ohmic layer results in a much lower contact resistance and a smooth contact surface morphology. The low ohmic contact resistance of 0.56 Ω mm has been achieved with moderate post-metal annealing settings of 600 °C, one of the lowest recorded values for similar metal schemes. Gao et al. [241] experimented with this idea by proposing a quadruple metal stack of Ti/Al/Ni/Ti ohmic contacts and showed an increased edge sharpness and surface metal morphology. This research also revealed an increased breakdown voltage, a more concentrated statistical distribution, and a lower ohmic contact resistance.

Constant et al. [242] investigated barrier height dependence on specific contact resistance for Au-free ohmic contacts generated on AlGaN/GaN heterostructures. They found that lowering the AlGaN thickness to an optimum level, at which a maximum polarization field-induced carrier density (ND-2DEG) is produced, reduces specific contact resistance. Li et al. [240] made a similar observation, discussing the reduced thickness of the barrier and the broader area for tunneling. Electron tunneling is therefore projected to improve massively, substantially lowering the contact resistance. On the other hand, it is also believed that the annealing temperature and ohmic groove etching significantly influence the ohmic behavior, device performance, and surface topography of HEMT devices [243]. Zhu et al. [244] have explored this theory by utilizing six different samples with varying combinations for the metal stack, as shown in Table 9. In sum, excellent ohmic connections are critical for regulating the annealing temperature and duration to balance the pace of different reactions.

Regarding metallization procedures with multi-layer Ti/Al structures and ohmic grooves, Zhu et al. [244] explained the detailed etching process, as depicted in Figure 21. The study highlighted the impact of different Ti/Al electrode layers, annealing temperatures, and ohmic groove depth on the ohmic characteristics of the HEMT devices. It found that the upgraded device achieved the best performance in terms of contact resistance, with the lowest specific contact resistivity of 2.2471 × 10^−5^ cm^2^ and the lowest contact resistance of 0.91014 Ω mm. These values are 71.8% and 54.3% lower than the conventional structure, respectively. These experimental results match older studies [243,245], with the Ti/Al layer for reduced resistance and the Ni/Au layer for smooth surface morphology, in which implementing an annealing temperature ohmic groove with rapid annealing at a high temperature is a decent approach to addressing the concern of high contact resistance. The duration of annealing is also an equally critical factor. In sum, excellent ohmic connections are essential for regulating the annealing temperature and duration in order to balance the pace of different reactions.

## 4. Conclusions and Future Prospects

In the past 6 years, we have seen the continuous development of AlGaN/GaN HEMT technology, and along the way, there have been many challenges encountered by designers to produce AlGaN/GaN HEMT with optimum performance. From a structural perspective, recent studies have proposed adequate solutions, showing opportunities for this technology to continue maturing. The following findings are drawn from the review:(1)As the industry demands normally-off devices for safety reasons, implementing the feature remains a research challenge. Ultimately, the issue requires solutions at the device technology level. p-GaN remains the only viable structural solution. However, issues such as degradation and mechanism failure still exist and require fixes to improve reliability and manufacturability. One suggestion for future improvement is to explore further the idea of a gate insulator layer that can achieve a high threshold voltage, a saturation drain current, and a turn-on voltage while reducing the gate leakage current and instability. This implementation involves the deposition of the Al_2_O_3_/AlN gate insulator layer in the p-GaN HEMT design, which can be further improved with modifications such as barrier layer variation. Nevertheless, more investigation is required before it can be widely adopted.(2)To address self-heating issues, comprehensive device thermal management, mainly focusing on the variation of extrinsic substrates as heat spreaders, is essential for reliable and robust HEMT devices. However, materials with a high thermal conductivity, such as diamond substrates, are still not a viable option due to their lack of compatibility with other substrate materials (GaN-diamond lattice mismatch). This leaves SiC as the most feasible option. To further close the gap between cost and performance, we propose the idea of exploring an LRSiC substrate. On top of being three times less expensive than a standard SiC substrate, it delivers better thermal management than a Si substrate. This suggests that LRSiC could be an excellent and cost-effective solution to the heat problem. However, more research is also required before it can be widely accepted.(3)Several structural solutions have been identified to resolve the challenges related to a high peak electric field, leakage current, and current collapse. To date, FP technology is a proven solution that can effectively control electric field distribution and lower the peak electric field below the GaN material’s critical electric field. Another possible way is through various structural modifications, including surface passivation, notches, trenches, gate structures, and barrier layer variations. Combining these different structures could resolve these issues. For instance, adding FP and notch structures on the same device could further reduce the challenges of a high peak electric field, leakage current, and current collapse.(4)Using different metallization strategies is a popular method of overcoming the problem of high-resistance ohmic contact. Researchers have demonstrated that implementing a stack of several materials may help improve ohmic contact resistance. We can expect other material combinations to be exploited in the near future, which may further enhance the ohmic contact.

Although tremendous improvements have been made in GaN device performance, there are still significant gaps between the observed device performance in real-world ap-plications and theoretical predictions. For example, cost and material crystallization quality must be considered in practical research, as these factors will eventually decide whether the ideas can be fulfilled commercially. This study has thus identified many structural conceptualizations proven to overcome existing challenges associated with HEMT devices. Ideal characteristics for practical transistor applications, such as a stable threshold voltage, a low leakage current, a high transconductance, an effective current control with a high linearity, and a wide dynamic input voltage range, have the potential to be discovered.

## Figures and Tables

**Figure 1 micromachines-13-02133-f001:**
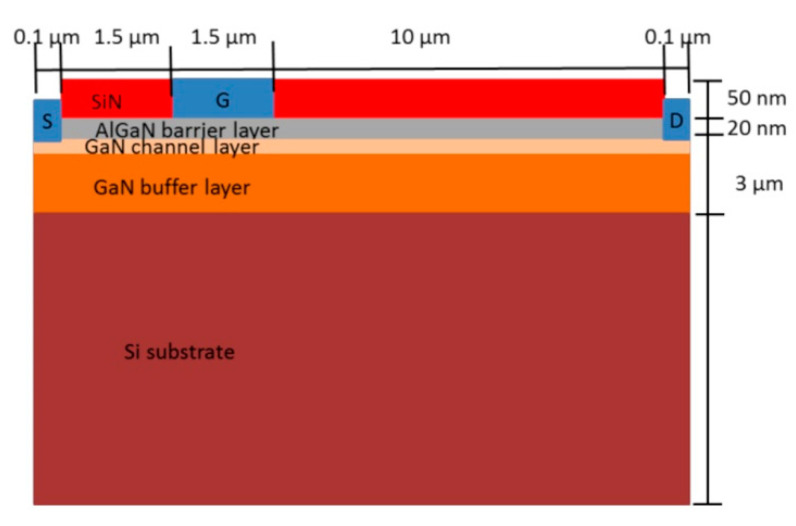
Schematic of a conventional AlGaN/GaN HEMT with an undoped AlGaN/GaN barrier layer, a substrate, a passivation layer, and metal contacts. The AlGaN layer aids in the polarization of the GaN region, which causes an oversupply of free dynamic electrons in the GaN layer. Reprinted with permission from ref. [42]. Copyright 2021 MDPI.

**Figure 2 micromachines-13-02133-f002:**
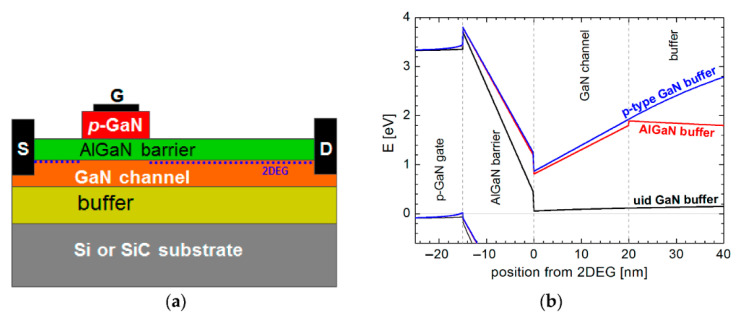
(**a**) A normally-off GaN transistor with a p-type doped GaN beneath the gate and (**b**) an energy band diagram showing the comparison between an AlGaN buffer, a p-type (compensation doped) GaN buffer, and an unintentionally doped GaN buffer. The channel region’s conduction band is displaced above the Fermi level. Reproduced with permission from ref. [72]. Copyright 2017 MDPI.

**Figure 3 micromachines-13-02133-f003:**
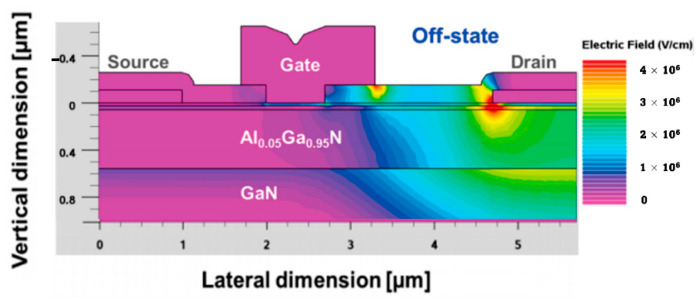
Electric field simulation in an off-state GaN HEMT with a drain bias of 300 V. Reprinted with permission from ref. [72]. Copyright 2017 MDPI.

**Figure 4 micromachines-13-02133-f004:**
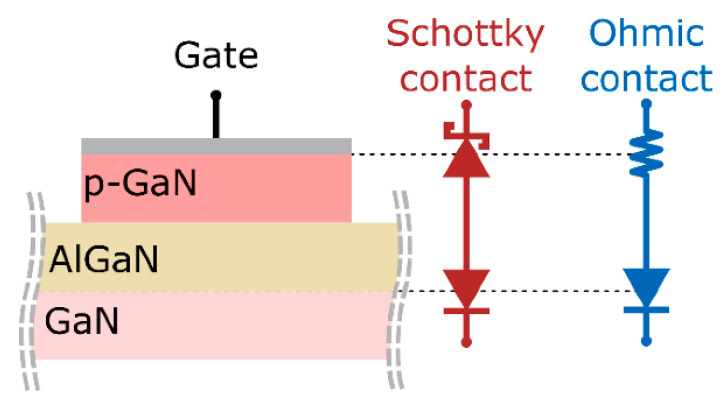
An ideal depiction of the cross-section of the gate heterostructure in p-GaN Gate HEMT. Reprinted with permission from ref. [78]. Copyright 2021 MDPI.

**Figure 5 micromachines-13-02133-f005:**
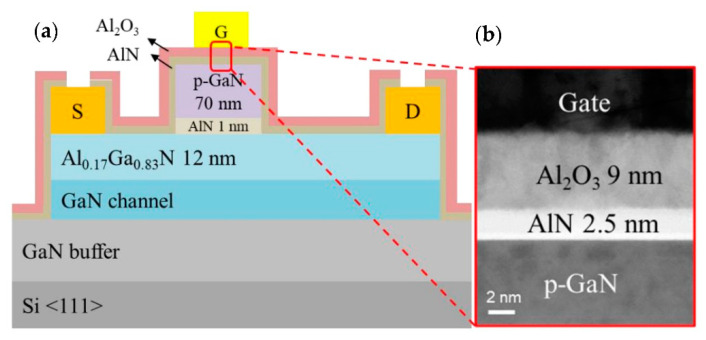
(**a**) Schematic device structure with L_GS_/L_G_/L_GD_/W_G_ = 2/4/10/100 µm and (**b**) transmission electron microscopy (TEM) image of the device. Reprinted with permission from ref. [97]. Copyright 2021 MDPI.

**Figure 6 micromachines-13-02133-f006:**
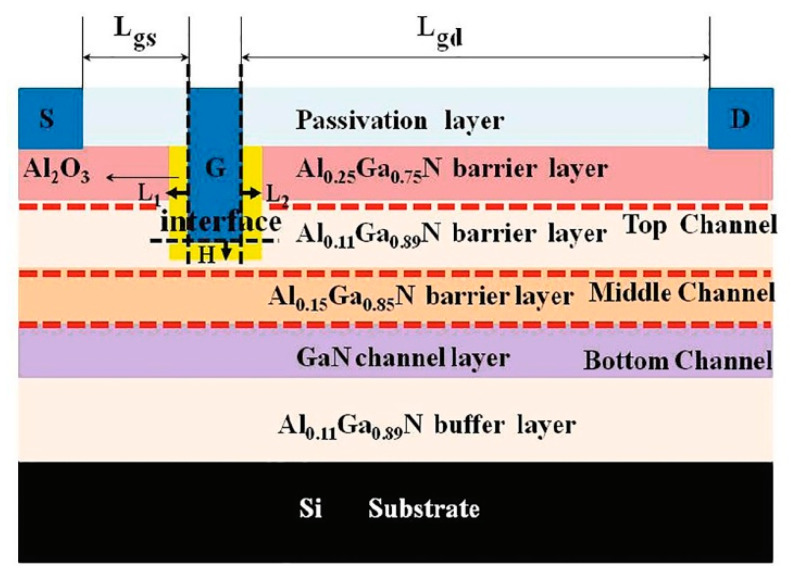
Schematic of an enhancement-mode HEMT with triple AlGaN barrier. Reprinted with permission from ref. [103]. Copyright 2021 Elsevier.

**Figure 7 micromachines-13-02133-f007:**
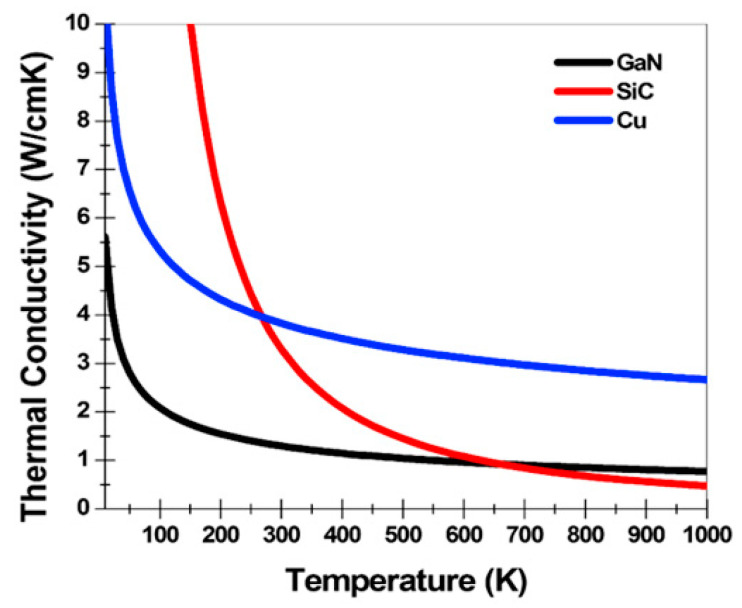
Thermal conductivity profiles for GaN, SiC, and Cu. At the highest recorded temperature of 1000 K, SiC has the lowest thermal conductivity, followed by GaN and Cu. Reprinted with permission from ref. [137]. Copyright 2020 MDPI.

**Figure 8 micromachines-13-02133-f008:**
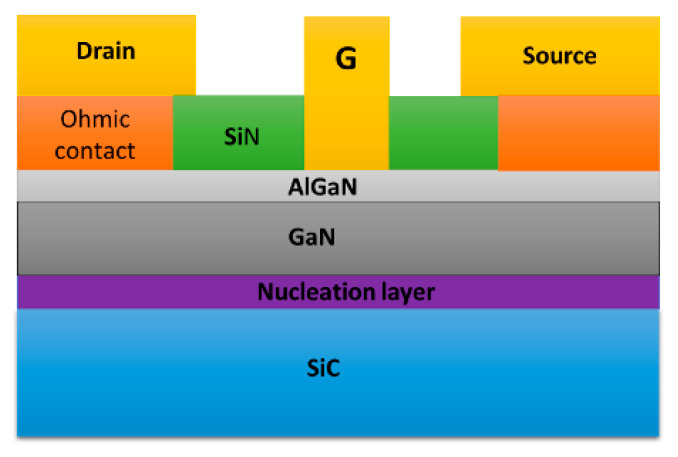
HEMT structure with SiC substrates. The nucleation layer, also known as the thermal boundary resistance (TBR), is used to achieve mesh tuning for the SiC and GaN layers and decrease mechanical stress and flaws in the GaN layer. The upper half of the GaN layer includes the 2DEG, and the AlGaN layer forms a heterojunction with the GaN layer. Reprinted with permission from ref. [123]. Copyright 2021 MDPI.

**Figure 9 micromachines-13-02133-f009:**
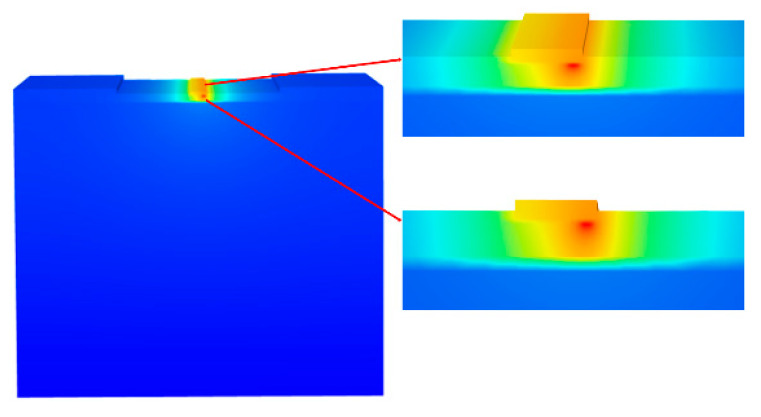
Temperature distribution in the HEMT grown on a SiC structure. As shown, in the AlGaN layer, the temperature is high near the gate, around the passivation layer, and along the gate. Reproduced with permission from ref. [123]. Copyright 2021 MDPI.

**Figure 10 micromachines-13-02133-f010:**
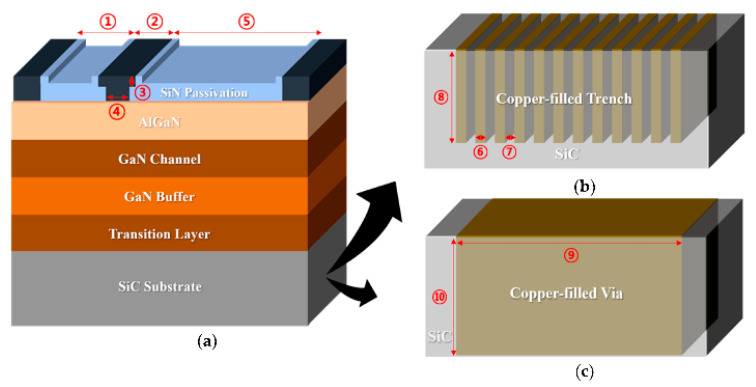
Structures of (**a**) AlGaN/GaN HEMT on SiC, (**b**) Cu-filled thermal trench, and (**c**) CTV in SiC substrate. The lattice temperature within the 2DEG channel was lowered, thus improving breakdown voltage, saturation current, drain current, and peak transconductance. Reprinted with permission from ref. [137]. Copyright 2020 MDPI.

**Figure 11 micromachines-13-02133-f011:**
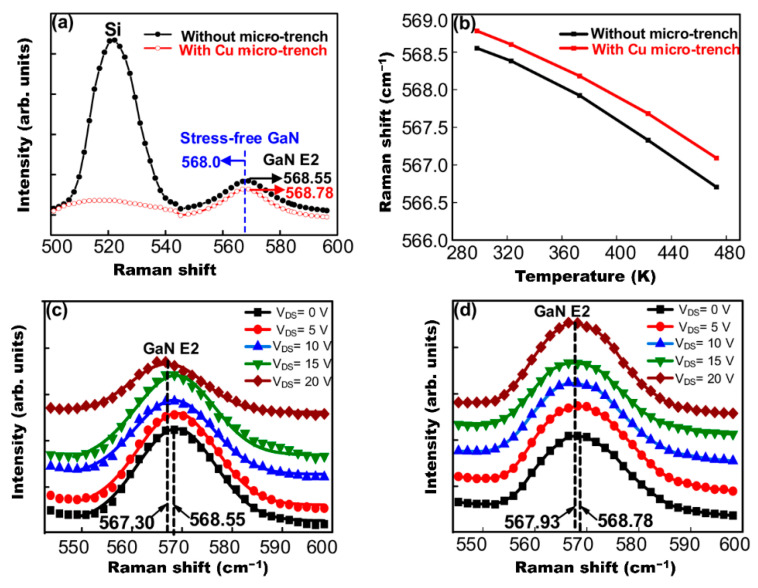
Validation of temperature rise in a device structure with Raman thermometry: (**a**) peak frequency of GaN and Si E2, (**b**) dependence on temperature of the active E2 mode in GaN, (**c**) peak E2 shift of GaN without micro-trench fabrication, and (**d**) peak E2 shift using a Cu-filled micro-trench, which is lower than without the micro-trench. Based on the channel temperature at various drain-source biases (V_ds_) in the Cu-deposited trench structure, the shift of the E2 (high) peak is reduced, suggesting effective heat removal. Reprinted under terms of the CC-BY license [162]. Copyright 2019, Mohanty et al., published by Springer Nature.

**Figure 12 micromachines-13-02133-f012:**
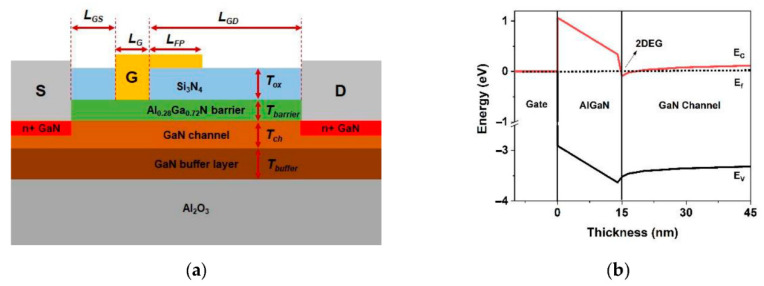
(**a**) View of AlGaN/GaN HEMT with FP structure and (**b**) equivalent energy band profile. The architecture allows for breakdown voltage improvement while preventing current collapse. Reprinted with permission from ref. [19]. Copyright 2021 MDPI.

**Figure 13 micromachines-13-02133-f013:**
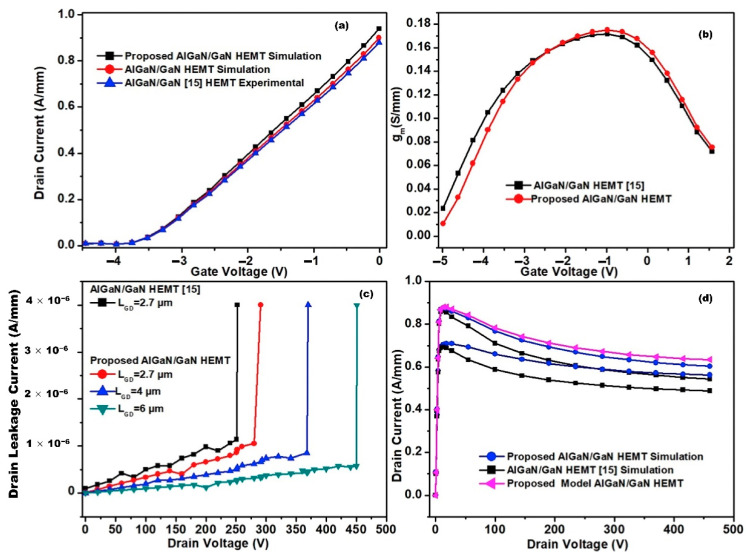
(**a**) Transfer characteristics, (**b**) g_m_ of AlGaN/GaN HEMT, (**c**) drain leakage current (I_ds_) versus V_ds_ at V_gs_ = −10 V, and (**d**) I_ds_–V_ds_ characteristics at V_gs_ = −1 V, 0 V. Reproduced with permission from ref. [184]. Copyright 2018 Elsevier.

**Figure 14 micromachines-13-02133-f014:**
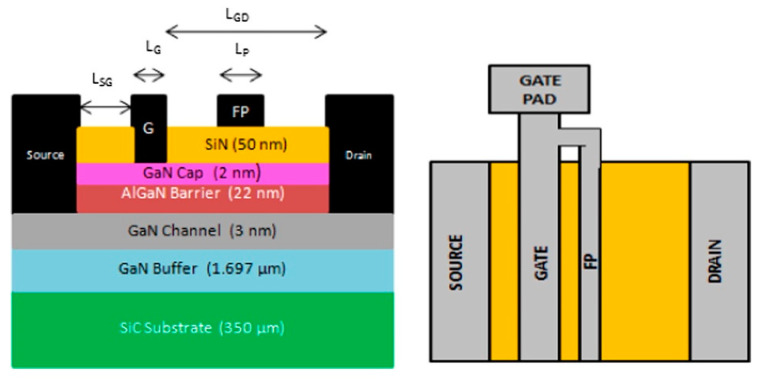
Cross-sectional and top views of an AlGaN/GaN HEMT structure with a discrete FP. Reprinted with permission from ref. [179]. Copyright 2019 Elsevier.

**Figure 15 micromachines-13-02133-f015:**
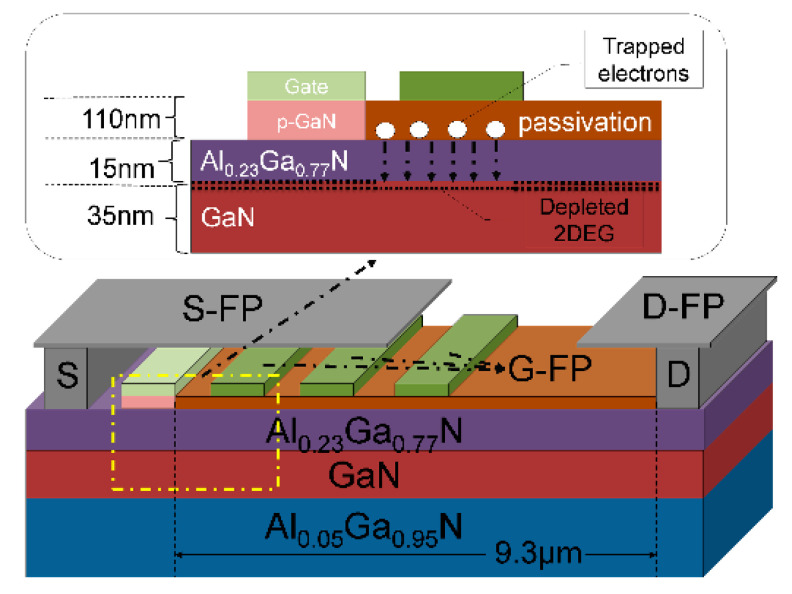
HEMT with a micro field plate (micro-FP) structure. Reprinted with permission from ref. [198]. Copyright 2021 MDPI.

**Figure 16 micromachines-13-02133-f016:**
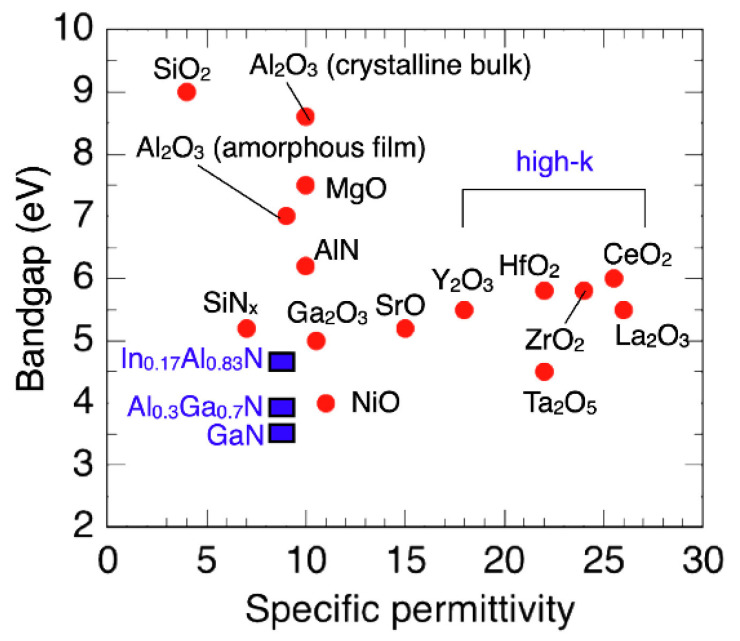
Band gaps and permittivity relationships for key insulators and GaN-based materials. Reprinted with permission from ref. [211]. Copyright 2018 Elsevier.

**Figure 17 micromachines-13-02133-f017:**
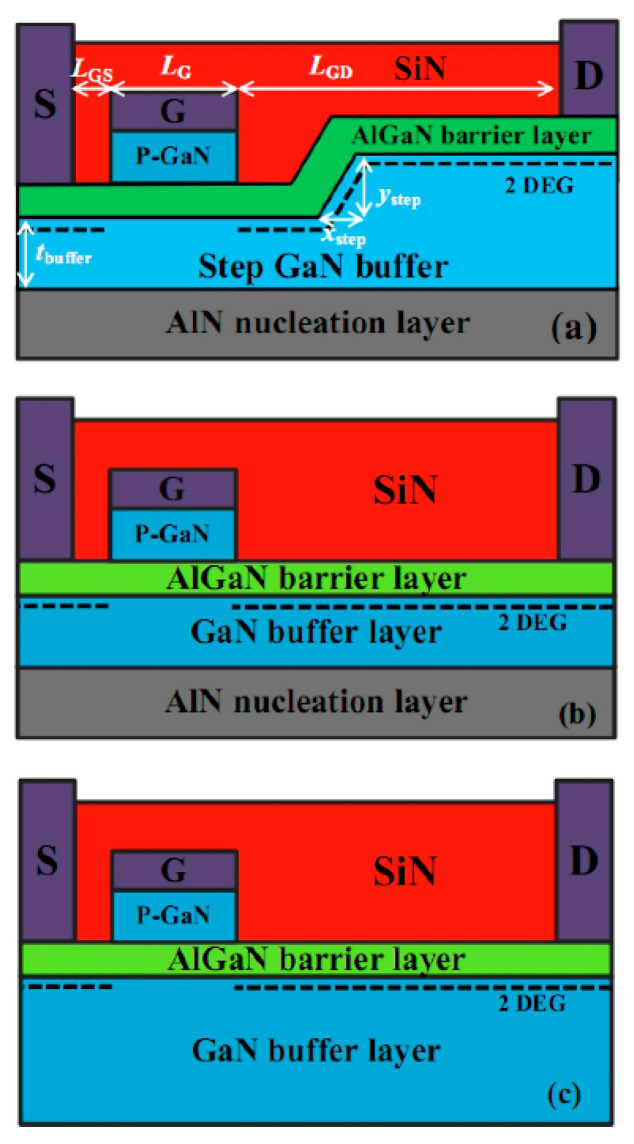
Proposed schematics of (**a**) SGB, (**b**) TGB, and (**c**) CGB. Reprinted with permission from ref. [219]. Copyright 2021 Elsevier.

**Figure 18 micromachines-13-02133-f018:**
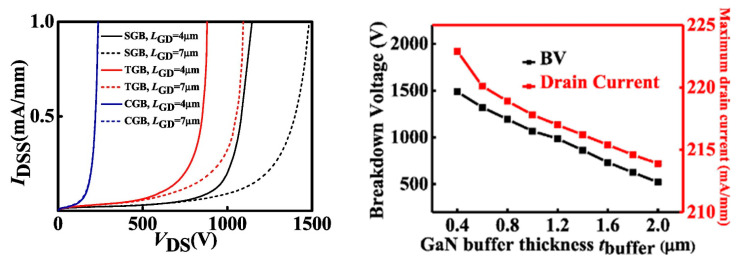
Breakdown voltage (V_BR_) of SGB, TGB, and CGB for different gate-to-drain distances (L_GD_), and dependence of V_BR_ and peak drain current on GaN buffer thickness. The developed step-etched GaN structure may alter the electric field distribution, resulting in a greater V_BR_ since the equipotential lines are more uniform than for standard HEMTs. Furthermore, at the ideal angle of the GaN buffer, the electron density at the AlGaN/GaN interface is unaffected, resulting in an output I_ds_–V_ds_ curve and current drive capacity comparable to conventional HEMTs. Reprinted with permission from ref. [219]. Copyright 2021 Elsevier.

**Figure 19 micromachines-13-02133-f019:**
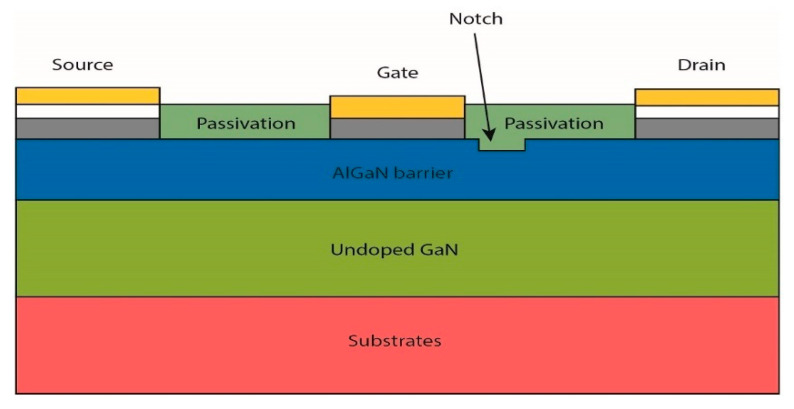
Illustration of AlGaN/GaN/HEMT structure with a notch formed between the passivation layer and AlGaN barrier.

**Figure 20 micromachines-13-02133-f020:**
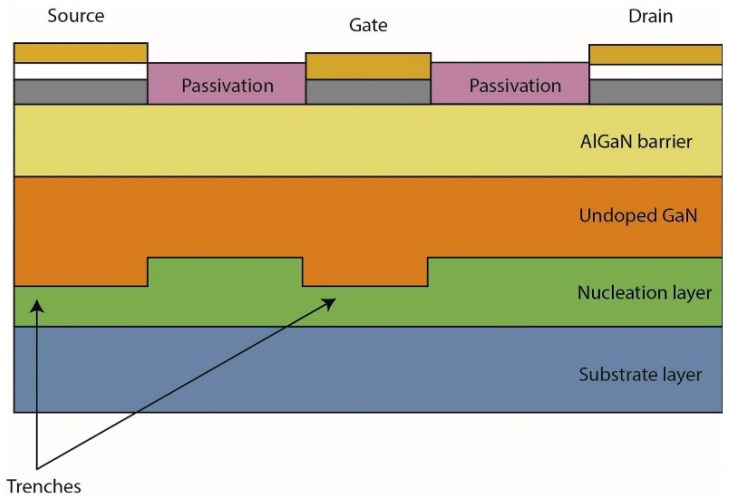
Illustration of a GaN-based HEMT with dual trench structures formed between the undoped GaN and the nucleation layer.

**Figure 21 micromachines-13-02133-f021:**
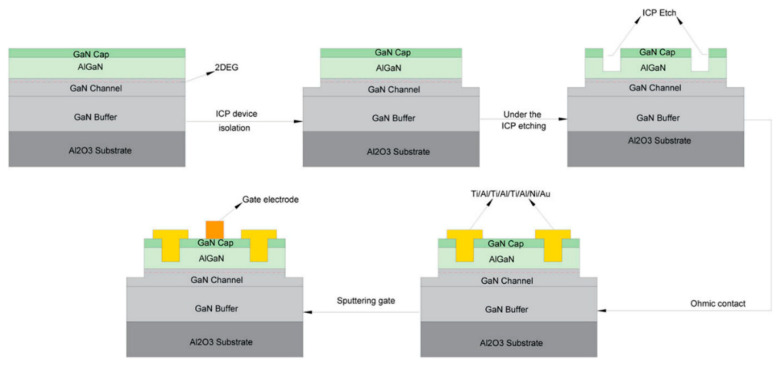
Process flowchart for device etching with an ohmic groove in Ti/Al/Ti/Al/Ti/Al/Ti/Al/Ni/Au. The procedure implemented the inductively coupled plasma (ICP) method to achieve device isolation, followed by rapid annealing at 850 °C for 30 s to form the ohmic contacts and then the final deposition of Ni/Au electrodes. Reproduced under terms of the CC-BY license [244]. Copyright 2021, Zhu et al., published by AIP Publishing.

**Table 1 micromachines-13-02133-t001:** Material parameters of GaN/AlGaN devices.

Parameters	Unit	Al_x_Ga_1-x_N	GaN
Electron mobility (µ_n_)	cm^2^ V^−1^ s^−1^	300.0–985.5	800–1350
Hole mobility (µ_p_)	cm^2^ V^−1^ s^−1^	10.0–13.3	10.0–22.0
Energy band gap (E_g_)	eV	3.87–5.10	3.299–3.550
Conduction band density of state (N_c_)	10^18^ cm^−3^	2.07–2.75	1.07–2.24
Valance band density of state (N_v_)	10^19^ cm^−3^	1.16–2.06	1.16–2.51
Electron affinity (χ)	eV	3.41–4.01	4.00–4.31
Saturation velocity (V_sat_)	10^7^ cm s^−1^	0.27–1.50	1.90–2.70
Relative permittivity (ɛ)	-	8.79–9.55	8.90–9.70
Al mole concentration (x)	-	0.26–0.85	-

**Table 2 micromachines-13-02133-t002:** Data for normally-off HEMTs with p-GaN gate, from recent literature.

Metal Gate	Structural Parameters	p-GaN Thickness (nm)	p-GaN Doping (cm^−3^)	V_th_ (V)	V_BR_ (V)	g_m,max_ (mS mm^−1^)	R_on_ (Ω·mm)	I_ds,max_ (mA mm^−1^)	Ref.
Ti/Al/Ti	L_G_ = 2 μm, L_GD_ = 5 μm, L_GS_ = 2 μm	1400	-	2.8	-	-	-	400	[81]
Ti/Al	L_G_ = 20 μm, W_G_ = 2π⋅65 μm, L_SD_ = 40 μm	50	3 × 10^19^	1.5	-	-	-	-	[82]
Mo/Ni	L_G_ = 2.0 μm, L_GD_ = 10 μm, L_GS_ = 1.5 μm	80	3 × 10^19^	1.08	560	150.0	10.7	554	[83]
Ti/Au	W_G_ = 100 μm, L_G_ = 2 μm, L_GD_ = 15 μm, L_GS_ = 4 μm	70	-	1.2	-	-	-	-	[84]
Ni/Au	W_G_ = 60 mm, L_G_ = 4 μm, L_GD_ = 3 μm, L_GS_ = 3 μm	70	4 × 10^19^	2.2	-	-	43.6	112.5	[85]
Zr/Au	W_G_ = 100 μm, L_G_ = 3 μm, L_GD_ = 7 μm, L_GS_ = 2 μm	80	-	1.5	-	-	-	-	[86]
Ti/Au	W_G_ = 100 μm, L_G_ = 4 μm, L_GD_ = 6 μm, L_GS_ = 3 μm	85	5 × 10^19^	1.1	300	-	10.0	355	[87]
Ni/Au	W_G_ = 0.25 mm, L_G_ = 1.3 μm	95	3 × 10^19^	1.5	>800	-	-	300	[88]
Ni/Au	W_G_ = 2 × 150 μm, L_G_ = 1 μm, L_GD_ = 3 μm, L_GS_ = 1 μm	80	-	0.5	-	81.5	8.2	215.9	[89]
Ni/Au	W_G_ = 10 μm, L_G_ = 0.7 μm, L_GD_ = L_GS_ = 0.75 μm	70	1 × 10^20^	1.6	>10	-	-	-	[90]
Ni/Au	W_G_ = 100 μm, L_G_ = 3 μm, L_GD_ = 10 μm, L_GS_ = 5 μm	60	4 × 10^19^	1.6	-	68.0	23.0	153	[91]
Ni/Au	L_G_ = 3 μm, L_GD_ = 7 μm, L_GS_ = 2 μm	60	3 × 10^19^	2.1	218	-	5.65	272	[92]
Ti/Au	W_G_ = 100 μm, L_G_ = 5 μm, L_GD_ = 10 μm, L_GS_ = 3 μm	100	4 × 10^19^	1.86	12.05	-	12.8	-	[93]
Ni/Au	W_G_ = 2 μm, L_G_ = 4 μm, L_GD_ = 15 μm, L_GS_ = 2 μm	100	3 × 10^19^	1.4	740	-	11.0	-	[94]

**Table 3 micromachines-13-02133-t003:** Electrical characteristics of multiple-barrier GaN-based HEMTs, based on recent literature. The threshold voltage (V_th_) varied greatly depending on the AlGaN barrier layers’ design and recessed-gated depth (H).

Barrier Designs	I_on_/I_off_	SS (mV dec^−1^)	g_m,max_ (mS mm^−1^)	I_ds,max_ (A mm^−1^)	V_th_ (V)	Ref.
Al_0.3_Ga_0.7_N/Al_0.2_Ga_0.8_N/GaN (Remaining bottom of 5 nm Al_0.15_Ga_0.88_N)	4.8 × 10^10^	87.9	7.6	-	0.25	[101]
Al_0.3_Ga_0.7_N/Al_0.2_Ga_0.8_N/GaN (Remaining bottom of 3 nm Al_0.15_Ga_0.85_N)	5.5 × 10^10^	229.3	71.2	-	3.25	
Al_0.3_Ga_0.7_N/Al_0.2_Ga_0.8_N/GaN (Remaining bottom of 5 nm Al0.2Ga0.8N)	1.2 × 10^11^	80.7	39	2.9	~0	
Al_0.3_Ga_0.7_N/Al_0.2_Ga_0.8_N/GaN (Remaining bottom of 3 nm Al_0.2_Ga_0.8_N)	-	153.1	90	-	0.5	
Al_0.25_Ga_0.7_N/Al_0.11_Ga_0.89_N/Al_0.15_Ga_0.8_N (H = 6 nm)	-	-	94.4	1.06	0.99	[103]
Al_0.25_Ga_0.7_N/Al_0.11_Ga_0.89_N/Al_0.15_Ga_0.8_N (H = 2.06 nm)	-	-	61.1	0.95	2.06	
AlN/GaN/AlN/GaN	-	-	200	1.2	~−4	[104]
AlGaN/GaN/AlGaN/GaN (Fin-shaped structure)	-	-	245	~0.5	0.2	[105]

**Table 4 micromachines-13-02133-t004:** Comparison of LRSiC and High-Resistivity SiC (HRSiC) Substrates. Reproduced with permission from ref. [148]. Copyright 2021 MDPI.

	Reference Price (USD)	Resistivity (Ω·cm)
LRSiC (6 in)	1000	0.015~0.025
HRSiC (6 in)	300	1 × 10^−5^

**Table 5 micromachines-13-02133-t005:** Output GaN-based HEMT based on different properties of substrates/films. Reproduced with permission from ref. [151]. Copyright 2021 MDPI.

Parameters	I_D_ at V_G_ = 0.0 V (mA mm^−1^)		g_m,max_ (mS mm^−1^)	Max. µ	Max. µ (cm^2^ V^−1^ s^−1^)	f_T_ (GHz)	f_max_ (GHz)
	at V_D_ = 0 V	at V_D_ = 10.0 V					
Sapphire substrate	658	542	220	1109	408	14.8	28.6
AlN substrate	717	705	251	1189	393	16.3	31.1
Cu film	776	795	271	1253	389	16.6	32.6

**Table 6 micromachines-13-02133-t006:** Comparison of output characteristics of GaN-based HEMTs with FPs.

L_FP_ (µm)	L_G_ (µm)	L_GD_ (µm)	Passivation Layer	Cap Layer	f_T_ (GHz)	Electric Field (MV cm^−1^)	V_th_ (V)	g_m,max_ (mS mm^−1^)	I_ds,max_ (mA mm^−1^)	V_BR_ (V)	Ref.
0.67	0.26	2.00	SiN	-	41.00	0.71	0.65	780.0	1060	138	[173]
1.75	0.40	-	SiN HfO_2_	-	40.00 28.00	-	−4.30	434.8 434.0	2160 2110	872 912	[174]
0.60	0.70	-	SiN	-	-	-	-	-	-	150	[177]
0.90	0.25	2.70	SiN	GaN	20.00	17.00	-	270.0	760	330	[179]
2.00	3.00	22.00	-	-	-	3.00	-	-	-	2200	[183]
1.00	0.25	2.70	SiN	AlN/GaN	-	2.90	0.50	175.0	900	291	[184]
1.50	1.50	5.00	Si_3_N_4_	-	-	-	−4.00	70.0	310	970	[19]
2.00	2.00	15.00	SiN	GaN	-	-	−3.50	138.0	-	365	[185]
3.00	1.00	8.00	SiN	-	19.00	4.87	-	-	3400	376	[186]
0.30	0.30	1.50	SiN	-	-	2.70	−5.84	-	-	400	[187]
0.80	0.50–1.00	3.55	Si_3_N_4_	-	6.70	-	−2.683	58.0	~100	669	[188]
0.20	0.25	2.70	SiN	GaN	28.30	-	-	350.0	1000	254	[189]
0.10	0.25	2.70	SiN	GaN	28.00	1.80	-	314.0	820	342	[190]
0.50	0.25	2.70	SiN	GaN	47.07	-	−4.30	323.0	1080	298	[191]
0.75	0.2	1.35	SiN	-	62.40	-	−2.60	-	1000	140	[192]
0.30	0.23	1.00	SiCN	-	-	-	-	-	-	282	[193]
0.80	0.25	1.00	SiN	-	38.00	-	−3.30	58.7	-	127	[194]

**Table 7 micromachines-13-02133-t007:** Properties of different high-k materials. Reproduced under terms of the CC-BY license [204]. Copyright 2021, Babaya et al., published by Universitas Ahmad Dahlan (UAD).

Materials	ε (F m^−1^)	C (J K^−1^)	K (W m^−1^ k^−1^)	Energy Gap (eV)	Ec (eV)	Ev (eV)
SiO_2_	3.9	3.066	0.014	9	3.5	4.4
SiN	7.5	0.585	0.185	-	-	-
Al_2_O_3_	9.3	3.14	0.29	8.8	3	4.7
Hf0_2_	22	-	-	5.8	1.4	1.3
TaO_5_	26	-	-	-	-	-
TiO_2_	80	-	-	3.5	1.1	1.3

**Table 8 micromachines-13-02133-t008:** Main features of two commercial GaN-on-Si HEMT technologies with different gate lengths. Reproduced with permission from ref. [224]. Copyright 2021 MDPI.

Parameter	GaN Processes	
	D01GH	D006GH
Gate length	100 nm	60 nm
Cut-off frequency	110 GHz	150 GHz
Maximum oscillation frequency	180 GHz	190 GHz
Gate–drain breakdown voltage	36 V	36 V
Maximum drain current density	1200 mA mm^−1^	1200 mA mm^−1^
Maximum extrinsic transconductance	800 mS mm^−1^	950 mS mm^−1^
Minimum noise figure at 40 GHz	1.5 dB	1.1 dB
RF power density	4 W mm^−1^	3.3 W mm^−1^

**Table 9 micromachines-13-02133-t009:** Summary of data based on six samples annealed at 850 °C. Reproduced under terms of the CC-BY license [244]. Copyright 2021, Zhu et al., published by AIP Publishing.

Samples	A	B	C	D	E	F
Metal stack	Ti/Al/Ni/Au	Ti/Al/Ni/Au	Ti/Al/Ni/Au	Ti/Al/Ti/Al/ Ti/Al//Ni/Au	Ti/Al/Ti/Al/ Ti/Al//Ni/Au	Ti/Al/Ti/Al/ Ti/Al//Ni/Au
Etching depth (nm)	0	10	20	0	10	20
Annealing temperature (°C)	850	850	850	850	850	850
Annealing time (s)	30	30	30	30	30	30
R_C_ (Ω mm)	1.6748	1.1597	1.1535	1.6554	0.9101	1.0108
ρ_c_ (Ω cm^2^) × 10^−5^	7.9677	3.5520	3.6413	6.3174	2.2471	2.6838
rms (nm)	105	55.7	81.9	75.8	42.5	52.2
R_a_ (nm)	87.3	65.4	68.4	63.3	33.3	42.6

## Data Availability

Not applicable.

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
