# Peer review of "Challenges and Opportunities for High-Power and High-Frequency AlGaN/GaN High-Electron-Mobility Transistor (HEMT) Applications: A Review"

_micromachines, 2022, doi:10.3390/mi13122133_

Round 1

Reviewer 1 Report

This review paper covers a wide range of topics.  I feel this would be useful to researchers in the field. However, I would like to provide a few suggestions. 
Since the paper is more focused on AlGaN/GaN HEMT, I would suggest modifying the title accordingly.
Another suggestion is to add some discussion of the interface properties (page 21), especially interface states of the devices using high-κ dielectrics (i.e. high-k dielectric/AlGaN). A brief discussion would add to making this review paper more comprehensive.

Reviewer 2 Report

The paper presents the state of the art of GaN-based technology, indicating the challenges and opportunities concerning the reliability and performance of GaN devices considering the following issues: normally-on operation, self-heating effect, current collapse, electric field distribution, gate leakage, and Schottky and ohmic contact and others.
The authors did a tremendous amount of work and formulated many interesting conclusions and comments. The references have 231 items. So, the paper is also valuable because there is so much helpful information in one place. I think it would be an ideal paper to summarize a paper series, e.g. collected in a special edition. Otherwise, authors may expose to the accusation that the information in the paper is mostly well known. Therefore, this article should be included in a publication on GaN-based devices. It is my only doubt.
I have noticed a mistake in the reference numbering [79] in Tab. 2 and the text (Liu et al. [79]).
